# Blocking of P2X7r Reduces Mitochondrial Stress Induced by Alcohol and Electronic Cigarette Exposure in Brain Microvascular Endothelial Cells

**DOI:** 10.3390/antiox11071328

**Published:** 2022-07-06

**Authors:** Naveen Mekala, Nishi Gheewala, Slava Rom, Uma Sriram, Yuri Persidsky

**Affiliations:** Department of Pathology and Laboratory Medicine, Lewis Katz School of Medicine, Temple University, Philadelphia, PA 19140, USA; naveen.mekala@temple.edu (N.M.); nishi.gheewala@temple.edu (N.G.); slava.rom@temple.edu (S.R.); uma.sriram@temple.edu (U.S.)

**Keywords:** P2X7r, mitochondrial stress, ethanol, electronic cigarette, extracellular ATP, Ca^2+^ signaling in ER, brain endothelial cells, P2X7r antagonist, A804598, TRPV1channel

## Abstract

Studies in both humans and animal models demonstrated that chronic alcohol/e-cigarette (e-Cig) exposure affects mitochondrial function and impairs barrier function in brain microvascular endothelial cells (BMVECs). Identification of the signaling pathways by which chronic alcohol/e-Cig exposure induces mitochondrial damage in BMVEC is vital for protection of the blood–brain barrier (BBB). To address the issue, we treated human BMVEC [hBMVECs (D3 cell-line)] with ethanol (ETH) [100 mM], acetaldehyde (ALD) [100 μM], or e-cigarette (e-Cig) [35 ng/mL of 1.8% or 0% nicotine] conditioned medium and showed reduced mitochondrial oxidative phosphorylation (OXPHOS) measured by a Seahorse analyzer. Seahorse data were further complemented with the expression of mitochondrial OXPHOS proteins detected by Western blots. We also observed cytosolic escape of ATP and its extracellular release due to the disruption of mitochondrial membrane potential caused by ETH, ALD, or 1.8% e-Cig exposure. Moreover ETH, ALD, or 1.8% e-Cig treatment resulted in elevated purinergic P2X7r and TRPV1 channel gene expression, measured using qPCR. We also demonstrated the protective role of P2X7r antagonist A804598 (10 μM) in restoring mitochondrial oxidative phosphorylation levels and preventing extracellular ATP release. In a BBB functional assay using trans-endothelial electrical resistance, we showed that blocking the P2X7r channel enhanced barrier function. In summary, we identified the potential common pathways of mitochondrial injury caused by ETH, ALD, and 1.8% e-Cig which allow new protective interventions. We are further investigating the potential link between P2X7 regulatory pathways and mitochondrial health.

## 1. Introduction

Polydrug abuse (including smoking, alcohol use disorder) presents a significant health problem [1,2]. During the last few years, electronic cigarettes (e-Cig) have become very popular especially among adolescent users [3]. Use of alcohol and tobacco smoke increases the risk of vascular cognitive impairment (VCI) since they affect endothelial cell functions in brain and other organs [4,5]. Onset of VCI can be attributed to blood–brain barrier (BBB) injury secondary to alterations in cytoskeleton and tight junctions (TJs), and proinflammatory damage [6,7]. BBB is an active vascular interface between the blood and central nervous system made of brain microvascular endothelial cells (BMVECs) connected by TJs in an impermeable monolayer [8,9]. Chronic drug abuse is known to stimulate the oxidative stress in BMVECs by overproduction reactive oxygen species (ROS) that disrupt the integrity of TJ and damage BBB [10,11]. Previous studies from our laboratory highlighted how oxidative stress created by methamphetamine, ETH, cigarettes, and e-Cig compromised the BBB [12,13,14]. Mitochondrial dysfunction has been suggested as a cause of enhanced ROS generation in the context of tobacco smoke and e-Cig exposure [4]. In case of alcohol abuse, elevated blood ETH levels are rapidly metabolized into toxic ALD and this toxic ALD accumulation is known to be very quick in endothelial cells, disrupting cell functions [15], and also causing mitochondrial dysfunction, ROS accumulation in case of Alzheimer’s [16]. 

Mitochondria are double membrane cell organelles which are very sensitive to endogenous ROS levels and increase in non-mitochondrial ROS due to pathophysiological conditions can augment mitochondrial ROS (mROS) [17]. This comprehensive mROS buildup minimizes mitochondrial oxidative phosphorylation (OXPHOS) levels [18], damages the mitochondrial DNA replication and destabilizes the mitochondrial membrane potential (MMP) [19,20]. Loss of MMP is a signal of bioenergetic stress and results in cytoplasmic release of apoptotic factors leading to cell death [21]. 

OXPHOS is a redox mechanism, where mitochondria convert high energy electron carriers (NADH and FADH_2_) into adenosine 5′-triphosphate (ATP), the usable form of chemical energy in the body. Although most ATP stays intracellular, under certain pathological conditions ATP is released into the extracellular space as a signaling molecule [22]. More importantly, the role of extracellular ATP in inflammation has been established recently [23] and ATP released from apoptotic cells functions as a ‘find me’ messenger for phagocytes to migrate to the damaged site [22]. Additionally, ATP also provides information about injury/inflammation in the surrounding cells/tissues via purinergic receptors, P2X and P2Y autocrine/paracrine purinergic signaling [24].

Among P2X receptors, P2X7 receptor (P2X7r) is of particular interest due to its association with neuroinflammation, BBB disruption, and most importantly its connection with mitochondrial function [25,26,27]. Harmful secondary metabolites of alcohol like acetaldehyde (ALD) and nicotine are also known to escalate oxidative stress in endothelial cells [14] and trigger mitochondrial dysfunction through activated TRPV1 ion-channels and P2X7r [28]. To explore the molecular mechanisms between drugs of abuse and P2X7r activation, we hypothesize that blocking P2X7r and TRPV1 ion-channels by a P2X7r antagonist could protect mitochondrial health.

In this report we treated hBMVECs with A804598 (10 μM), a novel and competitive antagonist of P2X7r that is known to have high affinity for blocking P2X7 receptors [29,30]. Our aim was to identify unifying mechanisms of brain endothelial injury caused by ETH, ALD, and e-Cig vape. We demonstrated that mitochondrial dysfunction was triggered by ETH, ALD, and 1.8% e-Cig and tested the therapeutic effects of P2X7r inhibition in preventing hBMVECs injury. 

## 2. Materials and Methods

### 2.1. Reagents/Kits

ETH 200 Proof (Cat. No. 2716) was procured from Decon laboratories Inc. (King of Prussia, PA, USA), 99.5% pure ALD (Cat. No. 402788) was procured from ACROS organics, Geel, Belgium, and e-Cig (1.8% and 0% nicotine) from Pure E-Liquids (Peterborough, PE1 1SB, UK). P2X7r antagonist A804598 (Cat. No. 4473) was purchased from Tocris Bioscience (Bristol, UK) and dissolved in DMSO (Cat. No. D5879) from Sigma-Aldrich (St. Louis, MI, USA). Mitochondria/cytosol fraction kit was purchased from Abcam (Cat. No. ab65320, Cambridge, UK).

### 2.2. Cell Culture and Treatments

Human brain microvascular endothelial cells [hBMVECs (D3 cell-line)] provided by Dr Pierre-Olivier Couraud, Institut Cochin, INSERM U1016, CNRS UMR 8104, Université Paris Descartes (Paris, France) [31] were cultured using EBM-2 Basal Medium (Cat. No. CC-3156) supplemented with EGM-2 SingleQuots (Cat. No. CC-4176) from Lonza Biosciences (St. Bend, OR, USA). Depending upon the study, hBMVECs were grown in either 96-well culture plate, or 6-well plate, or 100-mm tissue culture plates coated with type-I collagen (Cat. No. 354236) from Corning. Confluent cells were later exposed to ETH (100 mM), ALD (100 μM), and e-Cig (35 ng/mL of 1.8% and 0% nicotine) [6] conditioned media with and without P2X7r antagonist A804598 (10 μM) pre-treatment (1 h). Cells treated with A804598 alone served as P2X7r antagonist control. Concentrations of the insults and A804598 used in this study were determined based on prior studies [14,32] and our preliminary dose response studies (data provided in Appendix A). Unless mentioned specifically, all experiments were carried out after overnight incubation after respective treatments.

### 2.3. Western Blot Analyses

Denatured proteins from hBMVECs were separated by SDS- polyacrylamide gels (4–12% tricine gels) and electroblotted onto nitrocellulose membranes. The membranes were blocked with Intercept blocking buffer (LI-COR, Lincoln, NE, USA) and incubated overnight with primary antibodies (dilution 1:1000), later probed with near-infrared secondary antibodies (LI-COR) (dilution 1:5000) and visualized with an Odyssey imaging system (LI-COR). The band intensities were quantified, and protein expression levels were calculated relative to either beta-actin or totals of target protein from the same membrane. The following primary anti-rabbit antibodies for p-IRE1α (3294T), p-ASK1 (3765S), p-P38 (4511T), p-GSK3β S9 (5558T), p-ERK1/2 (4370T), VDAC (4661T), BAX (2774S), HSP60 (12165T), and AIF-1 (4642S) were procured from Cell Signaling (Danvers, MA, USA). Primary anti-mouse OXPHOS antibody cocktail of complex I-V(ab110411), Bax inhibitor-1 [BI-1] (ab18852), t- GSK3β (ab93926), and anti-goat β-actin (Ab8229) were procured from Abcam (Cambridge, UK). 

### 2.4. Mito-Stress Test in Cultured D3 Cells

hBMVECs (30,000cells/well) were plated in seahorse XF96 microplate (Agilent Technologies, Inc., Santa Clara, CA, USA, further Agilent) and allowed to attach overnight. Four corner wells were left only with growth medium for background correction. Next day, cells were pretreated with A804598 (10 μM) for 1 h, before replacing with growth media conditioned with ETH, ALD, e-Cig (1.8% or 0% nicotine), with and without A804598 and incubated overnight. Next morning, mitochondrial stress spawned against our treatments was measured using Agilent “Seahorse cell mito-stress test kit” (cat#103015-100). In brief, growth media was carefully aspirated, and cells washed thrice in bicarbonate-free and phenol red-free DMEM (cat# 103680-100) supplemented with 10 mM glucose, 1 mM pyruvate, and 2 mM glutamine. Cell plate was incubated in a non-CO_2_ incubator at 37 °C until we calibrated the Seahorse cartridge. Upon calibration, XF96 cell culture plate was placed in the Seahorse analyzer and basal oxygen consumption rates (OCR) were measured. Later cells were sequentially challenged with 2.5 μM oligomycin (ATP synthase inhibitor), 0.5 μM FCCP (mitochondrial uncoupler), and 0.5 μM rotenone/antimycin A (complex I/III inhibitor, respectively) (all from Agilent Technologies (Santa Clara, CA, USA)) while mitochondrial respiration levels were continuously recorded. After the assay, the spare respiratory capacity was measured by subtracting the basal OCR from the maximal OCR.

### 2.5. Quantitative RT-PCR 

Total RNA was isolated using the TRIzol^TM^ method (Cat. No. 15596026) from Invitrogen (Thermo Fisher Scientific, Carlsbad, MA, USA). Total RNA (500 ng) was converted to complementary DNA (cDNA) using RT^2^ PreAMP cDNA Synthesis Kit (Cat. No. 330451, Qiagen, Germantown, MD, USA). TaqMan probes for human P2X7r (Cat# hs00175721), TRPV1 (Cat. No. hs00218912) and GAPDH (Cat. No. hs02786624) were procured from Thermo Fisher. cDNA was further probed by real-time qPCR using TaqMan Fast Advanced Master Mix (Cat. No. 4444557, Applied Biosystems, Thermo Fisher (Waltham, MA, USA)). All reactions were performed in triplicate, and relative expression levels of each gene were normalized with GAPDH. The relative foldchange of P2X7r and TRPV1 genes against the treatments was investigated using the delta-delta-Ct (ddCt) method and values were normalized to ddCt values of GAPDH. 

### 2.6. ATP Detection Assay

Luminescent ATP Detection Assay Kit from Abcam (Cat. No. ab113849, Cambridge, UK) was used to measure the extra cellular ATP (eATP) levels in hBMVECs. This experiment was carried out in a 96-well culture plate, where 30,000 cells/well were seeded and allowed to attach overnight. Cells were pretreated with A804598 (10 μM) for 1 h before exposing to ETH, ALD, and e-Cig conditioned media. After overnight incubation, 50 μL of the spent media was collected into a new 96-well plate and ATP levels were measured following manufacturer’s instructions.

### 2.7. Intracellular Calcium (Ca^2+^) Assay

hBMVECs pre-treated with or without A804598 were stimulated with ETH, ALD, and e-Cig conditioned media for 2 h, and cell pellet (2 × 10^6^) was snap frozen in liquid nitrogen. Intracellular calcium (Ca^2+^) levels were then determined using calcium assay kit from Abcam, (Cat. No. ab102505, Cambridge, UK) following the manufacturer’s instructions.

### 2.8. Barrier Function Measurement by Trans-Endothelial Electrical Resistance (TEER)

Measurement of TEER on hBMVEC monolayers was used to determine the effects of P2X7r inhibitor A804598 on barrier integrity. The TEER assay was performed using the 1600R electric cell-substrate impedances system (ECIS) (Applied Biophysics, Troy, NY, USA) as previously described [33,34]. The ECIS system provides real-time monitoring of changes in TEER. In brief, hBMVECs at 30,000 cells/well were plated on collagen type I coated 96W20idf electrode arrays (Applied Biophysics). Cells were allowed to form monolayers until stable TEER values were reached. After 6 days (with media change every three days), the monolayers were exposed to ETH, ALD, or 1.8% e-Cig after pretreating with A804598 (10 μM) for 2 h. Readings were acquired continuously at 4000 Hz at for 24 h. Data is presented as percent change from baseline TEER. Three independent experiments were performed with 3–5 replicates for each condition.

### 2.9. Statistical Analysis

Statistical data analysis was carried out using GraphPad Prism 9 software (GraphPad Software, Inc., San Diego, CA, USA). Statistical significance between experimental groups was performed using one-way ANOVA or two-way ANOVA (TEER data) and significance levels were set at *p* < 0.05.

## 3. Results

### 3.1. P2X7r Inhibition Ameliorated Mitochondrial Dysfunction in hBMVECs Exposed with ETH, ALD and 1.8% e-Cig

It is known that exposure to alcohol and cigarettes (tobacco smoke) results in tissue injury associated with oxidative stress [35]. It has been suggested that e-Cig could also produce similar effects [4]. Yet, the mechanisms of oxidative stress linked to mitochondrial dysfunction/stress have not been fully elucidated. In light of this we measured mitochondrial respiratory parameters include the spare respiratory capacity (SRC) or reserve respiratory capacity (RRC). In our study mitochondrial SRC/RRC represents the additional ATP that can be produced during OXPHOS in case of sudden energy demand [36]. 

In the current study, we measured mitochondrial SRC of hBMVECs after overnight exposure to ETH, ALD, and 1.8% e-Cig conditioned media. Figure 1A depicts the mito-stress graphs for our studies. Whereas ETH stimulation resulted in 50% decline in spare respiration, ALD led to 75% decline, and e-Cig caused a 60% drop (Figure 1B). E-Cig extract with 0% nicotine had no impact on SRC levels in comparison to other insults. hBMVECs treated with P2X7r antagonist, A804598, preserved the mitochondrial function impaired by ETH, ALD, and e-Cig and largely maintained the SRC levels against the insults.

### 3.2. Reduced Expression of Mitochondrial OXPHOS Markers (Complexes II and IV) after ETH, ALD and 1.8% e-Cig Exposure Were Normalized by P2X7r Inhibition

Oxidative stress induced by ETH, ALD, and 1.8% e-Cig exposure significantly reduced the expression of mitochondrial OXPHOS markers in hBMVECs. Mitochondrial succinate dehydrogenase (SDH -complex II) is a vital enzyme that participates in both the Krebs cycle and the electron transport chain that catalyzes the oxygen consumption during aerobic respiration. Cytochrome *C* oxidase (COX- complex IV) is the terminal electron acceptor of the OXPHOS cycle that converts molecular oxygen into water molecules and translocate protons from mitochondrial matrix to the inter-membrane space, necessary for ATP generation. We analyzed expression of these two proteins using Western blot and found a 45% drop in Complex- II expression and a 30% reduction in Complex-IV expression in comparison with the non-treated control group (Figure 2). hBMVECs exposed with 0% nicotine e-Cig extract did not show any effect on OXPHOS expression. P2X7r inhibition with A804598 (A80) preserved the expression of mitochondrial markers as well as mitochondrial OXPHOS levels after exposure to ETH, ALD, or 1.8% e-Cig.

### 3.3. P2X7r Antagonist Prevented the Intracellular Ca^2+^ Accumulation Induced by ETH, ALD, and 1.8% e-Cig Exposure 

ROS and proinflammatory signals triggered by ETH and cigarettes smoke are known to control the movement of Ca^2+^ through calcium-permeable ion channels including TRPV1 and P2X7 receptors [14,32]. To understand this mechanism further, we exposed hBMVECs to ETH, ALD, and 1.8% e-Cig conditioned media and measured Ca^2+^ levels. We demonstrated significant increases in the intracellular Ca^2+^ levels against all our insults except 0% e-Cig. ETH and 1.8% e-Cig exposure led to a thousand-fold increase in intracellular Ca^2+^ levels (Figure 3A,C), whereas ALD led to fifteen hundred-fold the Ca^2+^ levels (Figure 3B). Cells treated with A804598 (via blocking both P2X7r and TRPV1 ion channels) brought down the intracellular Ca^2+^ levels to less than five hundred-fold, thus limiting the sustained Ca^2+^ flow into ER and mitochondrial matrix.

### 3.4. P2X7r Inhibition Diminished Endoplasmic Reticulum (ER) Stress Caused by ETH, ALD and 1.8% e-Cig in hBMVECs

Under physiologic conditions, Ca^2+^ levels in ER are tightly regulated for proper protein folding, whereas excess of Ca^2+^ results in unfolded protein response (UPR). Sustained UPR leads to chronic or irreversible ER stress triggering apoptosis via non-canonical pathway initiated by the phosphorylation of IRE1α It is the most conserved ER membrane protein that regulates cell survival/death. We showed that BMVECs exposed to ETH, ALD, or 1.8% e-Cig significantly augmented the phosphorylation of IRE1α, and apoptosis signal-regulating kinase 1 (ASK1)-MAP3K, a pro-apoptotic downstream regulator of pIRE1α (Figure 4A). In parallel, we also found reduced anti-apoptotic BAX inhibitor-1 (BI-1) protein levels conserved in the ER. ER stress induced MAP3K activation further reduced the phosphorylation of anti-apoptotic proteins including GSK3β S9 and ERK1/2 and increased the phosphorylation of pro-apoptotic P38 levels (Figure 4B). By reducing the Ca^2+^ generated ER stress, 10 μM A804598 (A80) moderates the activation of MAP kinases and protects the cells.

### 3.5. P2X7r Inhibition Protected the Mitochondrial Function and Prevented Upregulation of Pro-Apoptotic Factors in Response to ETH, ALD, and 1.8% e-Cig Exposure

In response to various stimuli, surplus of Ca^2+^ in ER lumen opens additional ‘open Ca^2+^ channels’ between ER and the mitochondrial outer membrane increasing Ca^2+^ release into the mitochondrial matrix [37]. In such a scenario, expression of voltage-dependent anion channel (VDAC) is known to increase, as VDAC is a major component of the ‘open Ca^2+^ channels’ [38]. Proteins isolated from hBMVECs exposed to ETH, ALD, and 1.8% e-Cig showed increased VDAC expression (Figure 5A). Modest increase of Ca^2+^ in mitochondrial matrix is known to stimulate oxidative phosphorylation, but beyond a critical threshold the mitochondrial matrix calcium (Ca^2+^_m_) transmits and amplifies apoptotic signals [39]. Western blots of mitochondrial fractions revealed mitochondrial translocation of BAX protein leading to the loss of mitochondrial outer membrane, and damage to mitochondrial membrane potential (MMP) (Figure 5B). Compromised mitochondrial membranes result in the cytosolic release of pro-apoptotic transcription factor, namely apoptosis inducing factor-1 (AIF1), and cytochrome *C* (Figure 5C). hBMVECs treated with A804598 (A80) presented a reduced BAX activation, retained the MMP and foiled the cytosolic release of apoptotic inducing factor-1 (AIF1), and cytochrome *C*.

### 3.6. Elevated P2X7r and TRPV1 Gene Expression by ETH, ALD, and 1.8% e-Cig Exposure Was Lowered by P2X7r Antagonist

Drugs of abuse including alcohol and nicotine are known to cause neuroinflammation via coactivation of P2X7r and its associate TRPV1 ion channels while upregulation of these receptor channels is known to cause mitochondrial dysfunction by disrupting mitochondrial membrane potential [40,41,42]. Using qPCR, we observed a four-fold increase in P2X7r expression after ETH, ALD, and 1.8% e-Cig (Figure 6A). Expression of TRPV1 channel paralleled P2X7r showed a four-fold increase after ETH and ALD, and six-fold increase after1.8% e-Cig exposure (Figure 6B). BMVECs pre-treated with P2X7r antagonist A804598 significantly reduced the expression of both the P2X7r and TRPV1 channels. Expression of the P2X7r and TRPV1 channels remained unchanged with 0% e-Cig exposure.

### 3.7. A804598 Blocks the Extracellular ATP (eATP) Release

To understand the role of addictive drugs in ATP release, we quantified extracellular ATP (eATP) levels in hBMVECs stimulated with ETH, ALD, and 1.8% e-Cig conditioned media. After overnight exposure to ETH, loss in mitochondrial membrane potential resulted in a four-fold increase in eATP release, while both ALD and 1.8% e-Cig exposure showed an eight-fold increase in eATP concentration (Figure 7). hBMVECs pre-treated with P2X7r antagonist, A804598 presented very little to no change in eATP levels after ETH, ALD, or 1.8 % e-Cig media, and confirm the trail for eATP escape through purinergic receptor (P2X7r) and associated TRPV1 ion channels.

### 3.8. Inhibition of P2X7r Protects the Barrier Tightness against ETH, ALD, and 1.8% e-Cig Exposure in hBMVECs

To further validate the functional significance of P2X7r signaling on brain endothelial cells, which is barrier function, we used the TEER assay to assess if the endothelial resistance to the ETH, ALD, and 1.8% e-Cig insults can be rescued by blocking the P2X7r. Although effects of P2X7r inhibition on BBB has been indirectly demonstrated [43], direct effects on barrier function have never been shown. Further, no direct determination has been reported on the BBB enhancing effect using compound A804598. hBMVEC monolayers (plated on ECIS microelectrode arrays) were treated with P2X7r antagonist A804598, and TEER was monitored over the course of 24 h (Figure 8A,C,E). Resistance values in untreated cells (normalized to 100%) represent the steady state baseline TEER after confluent BMVEC monolayers were established and TJ was formed [44]. The time at addition of the stimulants was set at 0 h. Effects of the compounds to alter the resistance was observed starting at 12 h time point. Resistance values were significantly reduced by ETH, ALD, and 1.8% e-Cig (approximately 25% reduction) treatments. Addition of A804598 (10 μM) significantly increased the average TEER at 12 h for all treatments (Figure 8B,D,F). While the increased TEER pattern was obvious over the 24 h time-period (Figure 8A,C,E), the most significant changes were seen only at 12 h. The results overall suggest that blocking of P2X7r in brain endothelial cells may directly benefit barrier integrity.

## 4. Discussion

In this report, we demonstrated that alcohol and e-Cig induce mitochondrial dysfunction by similar mechanisms in human brain endothelial cells (D3 cells). Earlier studies have shown various pathways by which drugs of abuse impair the mitochondrial function due to mitochondrial DNA damage [45], electrons escaping through a broken electron transport system [46], through expression of pro-inflammatory cytokines like TNF-α or IL-1β [47], and phosphorylation of stress-activated MAPKs stimulating canonical mitophagy [48,49]. In our study, we showed a non-canonical pathway of mitochondrial regulation by the purinergic receptor, especially via P2X7r, using specific blocker A804598 (Please check the graphical abstract for experimental design). To our knowledge, this is the first report dissecting a mechanism of mitochondrial regulation by P2X7r and the associated TRPV1 channel in hBMVECs. 

Alcohol and e-Cig cause damage in the internal organs (liver, lungs, etc.) as well as the BBB and CNS [7,49,50]. ROS generated by tobacco smoke and e-Cig vape damage the brain endothelium causing cerebral ischemia and secondary brain damage [50]. Acute exposure to tobacco smoke and e-Cig also disables the expression of antioxidative Nrf2 proteins in the mouse brain endothelium resulting in BBB compromise [51]. Tobacco smoke is also known to promote mitochondrial depolarization in primary brain vascular endothelial cells [52]. In our prior studies, primary hBMVECs exposed to ETH or ALD demonstrated an increase in ROS and nitric oxide, resulting in TJ compromise and BBB injury [12,53]. Our recent in vivo experiments tested with e-Cig vape (0% or 1.8% nicotine) showed reduced expression of critical genes in the mouse brain, diminished the expression of occludin and glucose transporter 1 (Glut1) proteins, and increased BBB permeability, neuroinflammation, and cognitive impairment [6].

In recent years, the potential role of P2X7r in alcohol and nicotine-induced BBB damage has gained attention [54,55]. In our experiments, hBMVECs exposed to ETH, ALD, and 1.8% e-Cig reported a four-fold increase in P2X7r expression, and up to six-fold increase in TRPV1 levels, respectively. Up-regulation of P2X7r expression is an indication of the inflammasome pathway, promoting autocrine/paracrine signaling via ATP release [55]. Alongside P2X7r, TRPV1 channels also play a critical role in triggering inflammasome cascades, releasing proinflammatory cytokines damaging mitochondrial function and BBB integrity [27,46,56]. A804598 efficiently blocked the over expression of both P2X7r and TRPV1 channels pointing to the role of purinergic signaling in the autocrine regulation of receptor expression.

Prolonged activation of P2X7r and TRPV1 channels is known to create large non-selective membrane pores, allowing Ca^2+^ influx into the cytosol [57]. We showed previously that ETH and ALD stimulate Ca^2+^ release from ER [53]. Being a secondary messenger, excess cytosolic Ca^2+^ is toxic to the cell, inflicts cytoskeletal damage, and alters gene transcription in the cells [56,58]. To avoid this adverse situation, cells have developed a defensive mechanism where cytosolic Ca^2+^ is compelled into the ER through the SERCA pump, a Ca^2+^ ATPase that transfers Ca^2+^ from the cytosol to the ER lumen via ATP hydrolysis [59]. 

Although Ca^2+^ storage is one of the functions commonly attributed to the ER, a rapid and unrelenting flow of Ca^2+^ into ER lumen interrupts the protein folding in ER and powers the autophosphorylation of serine/threonine MAPKs (IRE1α, ERK1/2) and MAP3K (ASK-1), resulting in ER stress [60]. Once active, pASK-1 facilitates the cross talk between GSK-3β and p38 kinases and enhances the pro-apoptotic signals [61]. Simultaneously, excess Ca^2+^ levels in ER promote Ca^2+^ exchange/influx into the mitochondria, tethered to the ER. This Ca^2+^ exchange between mitochondria and ER is tightly regulated by the microdomains made of VDAC proteins [38,62]. In our experiments, hBMVECs exposed to ETH, ALD, and e-Cig presented over thousand-fold increase in intracellular Ca^2+^ levels and promoted the phosphorylation of ER stress-abetted proteins namely IRE1α, ASK-1, P38, and curtailed the phosphorylation of anti-apoptotic ERK1/2 and Bax Inhibitor-1 (BI-1) proteins. Inhibition of P2X7r in hBMVECs by A804598 reduced the intracellular Ca^2+^ levels by lowering the ER stress and prevented the activation pro-apoptotic MAPKs protecting the hBMVECs against ETH, ALD, and 1.8% e-Cig. Enhanced VDAC expression during ER stress also confirmed the crosstalk between mitochondria and ER.

Bax is a proapoptotic member of the Bcl-2 family, usually present in the cytosol in its inactive form. Upon activation, Bax penetrates the mitochondrial outer membrane (MOM) and disturbs the mitochondrial membrane potential (MMP) by creating pores in the MOM [63]. Once MMP is irreversibly damaged, it allows escape of apoptosis inducing factor-1 (AIF-1) and cytochrome *C* from the mitochondrial matrix into the cytosol and promotes cellular apoptosis [64,65,66]. Our western blot data from mitochondrial and cytosolic fractions confirmed the mitochondrial translocation of activated Bax after ETH, ALD, and e-Cig exposures. Simultaneously, we observed an increased amount of AIF-1 and cytochrome *C* escaped into cytosolic fractions. These observations confirm the similar mechanism destabilizing the MMP and backing mitochondrial dysfunction in hBMVECs due to ETH and 1.8% e-Cig. 

Optimal Ca^2+^ levels in mitochondrial matrix are known to manage the citric acid cycle by activation of pyruvate, isocitrate enzymes of the Krebs cycle [67]. Mitochondrial Ca^2+^ is also known to regulate ATP production through activation of mitochondrial complex-V (ATP synthase) [68]. That said, higher VDAC expression after ETH, ALD, and e-Cig exposure leads to a rapid influx of Ca^2+^ into the mitochondrial matrix, destabilizing Ca^2+^ homeostasis. Indeed, our Western blot data revealed a significant reduction in mitochondrial complex-II and complex-IV expression against the insults. This detrimental loss of mitochondrial complexes might be attributed to a combination of MMP loss and asymmetric Ca^2+^ levels accumulated in the mitochondrial matrix. P2X7r inhibition in hBMVECs restored the expression of mitochondrial complex- II and IV after exposure to ETH, ALD, or 1.8% e-Cig. 

In mitochondrial OXPHOS proteins, complex-II is a multimeric protein exclusively coded by the genomic DNA and later assembled into the inner mitochondrial membrane. Complex-II is known to accept electrons from FADH_2_ and transfer to complex III via CoQ10 [69]. Electrons are then carried by complex-III to complex-IV, and finally to molecular oxygen. This uninterrupted transport of electrons through the OXPHOS complexes creates a transmembrane proton gradient in the inner mitochondrial membrane, which is utilized by complex V, to convert ADP to ATP [70]. A defect in any of the above-mentioned complexes leads to electron seepage through the complexes and reduces mitochondrial function. In this context, our seahorse data corroborated with western blot results, where a significant depletion of mitochondrial complex-II and IV characterized a sharp decline in spare respiration after ETH, ALD, and 1.8% e-Cig treatments, and P2X7r antagonist restored the spare respiration levels in hBMVECs. 

Electron seepage through damaged mitochondrial complexes alters the mitochondrial redox state and promotes mitochondrial ROS (mROS) generation, and elevated mROS is central to the progression of multiple inflammatory responses [18,71]. Cells and tissues experiencing chronic toxic or inflammatory response send distress signals via release of eATP that bind to P2X7r and trigger a series of autocrine and paracrine signals [72]. In our studies, hBMVECs exposed to ETH showed a five-fold increase in eATP levels, whereas ALD and 1.8% e-Cig exposures yielded eight-fold increase in eATP levels. hBMVECs treated with P2X7r antagonist A804598 blocked the P2X7r and TRPV1 channels and diminished the eATP levels to control levels. These results strongly support the idea that P2X7r plays an important role in ETH, ALD, and 1.8% e-Cig induced oxidative stress.

TEER data of altered electrical resistance caused by ETH, ALD, and 1.8% e-Cig clearly demonstrate disruption of barrier function caused by these insults. A drop in TEER represents a decrease in barrier tightness that results from either change in expression or localization of TJ proteins (e.g., occludin, cl-5, or ZO-1) or in cytoskeletal reorganization due to activation of small GTPases [33,34,36]. Since TEER response peaked at 12 h and rebounded after 16 h to the baseline levels, this may be due to an acute reaction of BMVECs to the insult mostly through the effect on the cytoskeleton machinery and GTPase signaling. All these mechanisms will be explored in our future studies. Blocking the P2X7r by specific antagonist rescued this effect in vitro, suggesting an important role of P2X7 channel in BBB homeostasis. Further, functional barrier data support the idea that mitochondrial dysfunction, Ca^2+^ intracellular and ATP extracellular release underlie BBB injury by ETH or 1.8% e-Cig, and prevention of these negative effects by P2X7r inhibition preserves integrity. A recent study demonstrated that urothelial barrier function was normalized by P2X7r selective antagonist, A804598, in acrolein-induced bladder urothelial cell damage in a porcine model [30]. The multifaced role of P2X7r in BBB regulation during neuroinflammatory diseases has been recognized [73].

## 5. Conclusions

In summary, P2X7r and TRPV1 activation by ETH, ALD, and 1.8% e-Cig conditioned media greatly reduced the expression of mitochondrial OXPHOS proteins, and equally diminished the oxidative phosphorylation capacity in hBMVECs. By blocking P2X7r and TRPV1 channels, A804598 protected mitochondrial functions, prevented intracellular Ca^2+^ accumulation and extracellular ATP releases against ETH, ALD, or 1.8% e-Cig exposure. Further, our data indicate that alcohol, its metabolite ALD, and 1.8% e-Cig cause endothelial injury through the same pathway linked to caspase-independent pro-apoptotic pathways. Our findings point to P2X7r as a potential therapeutic target to prevent BBB damage triggered by alcohol or e-Cig vape. 

## Figures and Tables

**Figure 1 antioxidants-11-01328-f001:**
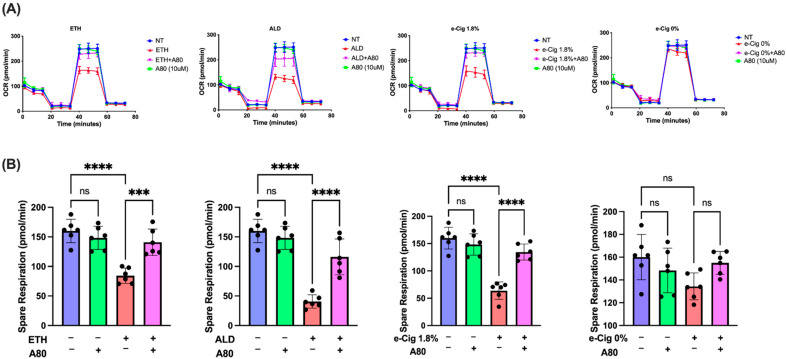
**Mitochondrial OXPHOS levels diminished by ETH, ALD or 1.8% e-Cig exposure were restored by P2X7r antagonist.** Mitochondrial SRC levels were measured after overnight exposure to ETH, ALD, and 1.8% e-Cig conditioned media. (**A**) shows the mito-stress graphs against ETH, ALD, and e-Cig (1.8%, 0% nicotine) stimulation. We quantified mitochondrial spare respiration (SRC) by subtracting basal respiration from the maximal respiration. (**B**) After overnight incubation, we recorded a 50% reduction in spare respiration against ETH exposure, 75% reduction against ALD, and 60% decrease after treatment with1.8% e-Cig containing media. Although SRC levels in cells treated with 0% e-Cig stimulus show a similar pattern, changes in SRC levels were not significant. Cells treated with P2X7r antagonist A804598 restored the SRC levels against ETH, ALD, and 1.8% e-Cig stimuli (*n* = 6). We normalized the Seahorse data to untreated control cells and One-Way ANOVA was used for statistical analysis, marked as *** *p* ≤ 0.001, **** *p* ≤ 0.0001 and ns (not significant).

**Figure 2 antioxidants-11-01328-f002:**
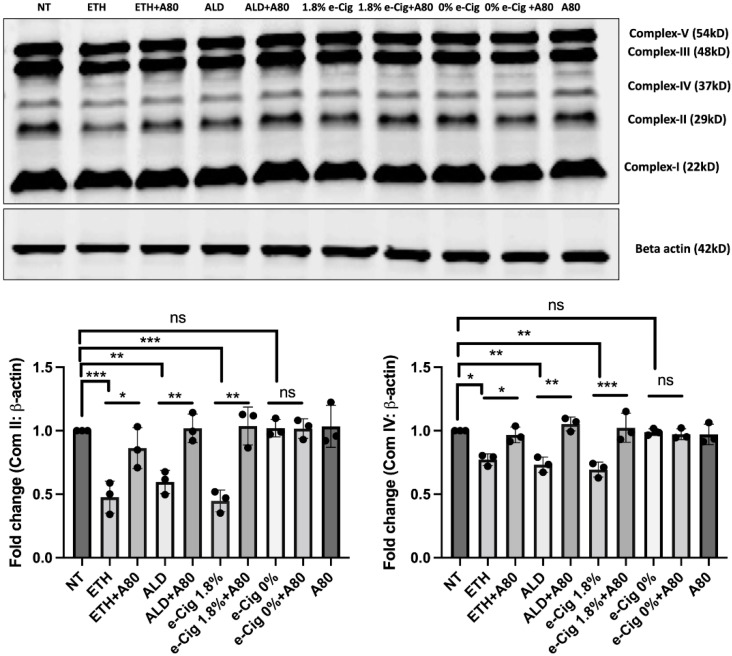
**P2X7r inhibitor A804598 protected the expression of mitochondrial OXPHOS proteins against ETH, ALD, or 1.8% e-Cig exposure.** When compared with control group hBMVECs incubated with ETH, ALD, and 1.8% e-Cig conditioned media reduced the expression of mitochondrial complex-II and complex-IV by 45% and 30%, respectively. Cells treated with A804598 (A80) preserved the expression of mitochondrial complexes (*n* = 3). Western blot data were normalized with β-actin and one-way ANOVA was used for the statistical analyses, marked as * *p* ≤ 0.05, ** *p* ≤ 0.01, *** *p* ≤ 0.001 and ns (not significant).

**Figure 3 antioxidants-11-01328-f003:**
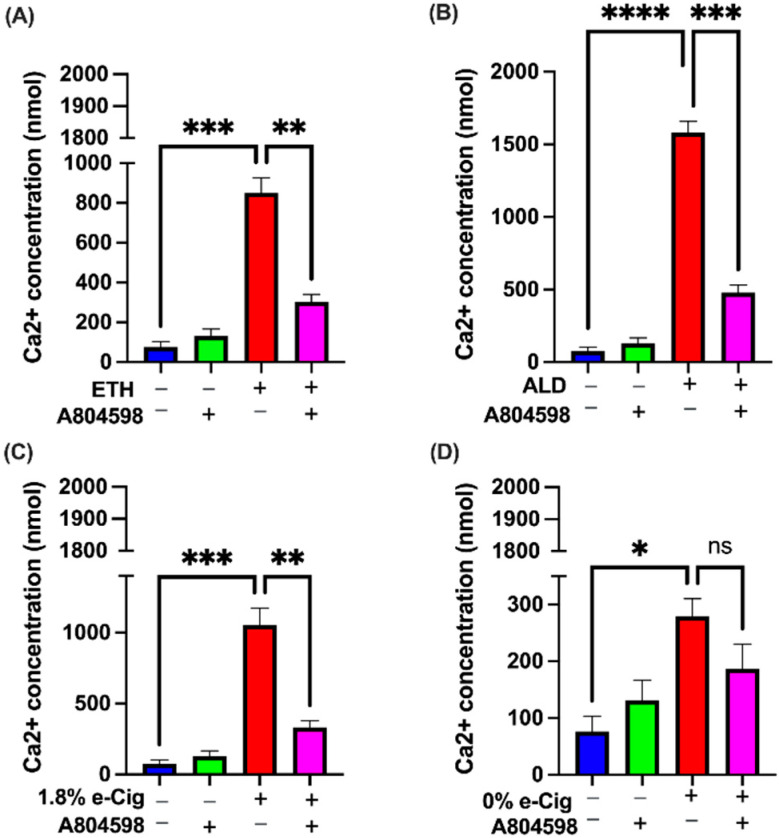
**By blocking P2X7r and TRPV1 channels, A804598 inhibited the intracellular Ca^2+^ influx enhanced by ETH, ALD, or 1.8% e-Cig**. hBMVECs exposed to (**A**) ETH, (**B**) ALD, and (**C**) 1.8% e-Cig showed one thousand-to-fifteen-hundred-fold increase in intracellular Ca^2+^ levels and cells treated with A804598 significantly diminished the intracellular Ca^2+^ levels. (**D**) e-Cig with 0% nicotine showed no impact on intracellular Ca^2+^ levels (*n* = 3). One-way ANOVA was used for the statistical analyses as * *p* ≤ 0.05, ** *p* ≤ 0.01, *** *p* ≤ 0.001, **** *p* ≤ 0.0001 and ns (not significant).

**Figure 4 antioxidants-11-01328-f004:**
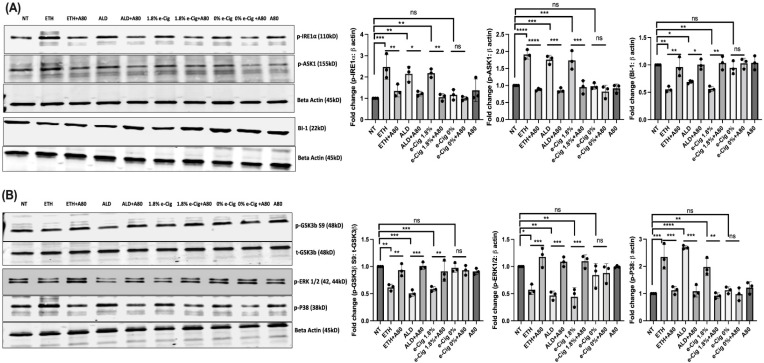
**P2X7r antagonist (A804598) reduced the non-canonical ER stress induced by ETH, ALD, and e-Cig stimulation in hBMVECs.** (**A**) ER Ca^2+^ buildup induced by ETH, ALD, and 1.8% e-Cig stimulus increased the expression pro-apoptotic p-IRE1α by three-fold and p-ASK1 by two-fold, respectively, and reduced the anti-apoptotic BAX inhibitor-1 (BI-1) protein expression by fifty percent (50%). (**B**) ER stress due to Ca^2+^ accumulation also significantly reduced the expression of p- GSK3β S9 and p-ERK1/2 and instigated p-P38 MAPK levels. hBMVECs treated with A804598 (A80) could prevent the ER stress induced apoptotic signals. Western blot data were normalized to β-actin/total GSK3β (*n* = 3). One-way ANOVA was used for the statistical analyses, marked as * *p* ≤ 0.05, ** *p* ≤ 0.01, *** *p* ≤ 0.001, **** *p* ≤ 0.0001 and ns (not significant).

**Figure 5 antioxidants-11-01328-f005:**
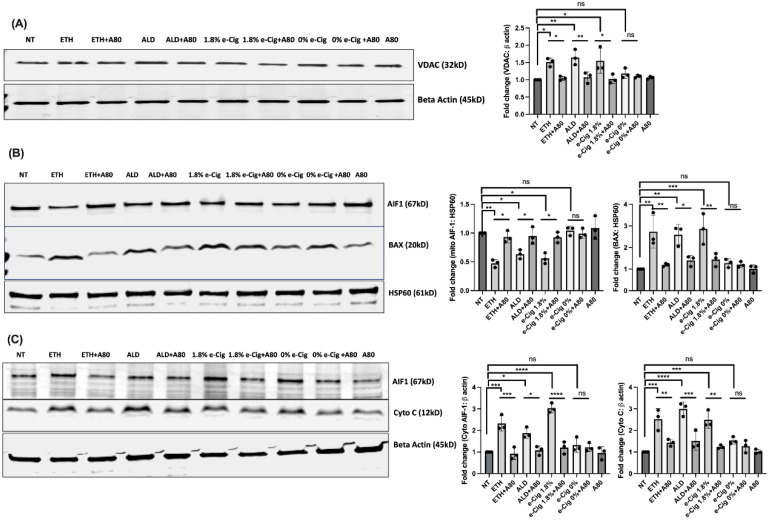
**P2X7r antagonist (A804598) protected the mitochondrial function against ETH, ALD, and 1.8% e-Cig stimuli.** (**A**) ETH stimulation increased the VDAC levels by fifty percent (50%), whereas ALD and 1.8% e-Cig stimulus doubled the VDAC expression. (**B**) Mitochondrial fractions from hBMVECs exposed to ETH, ALD, and 1.8% e-Cig showed a three-fold increase in BAX protein levels. Simultaneously, levels of AIF-1 proteins deteriorated by 50% with ETH, 55% after ALD, and up to 60% after 1.8% e-Cig exposure. (**C**) In parallel, cytosolic fractions presented greater AIF-1, and cytochrome C levels in response to the insults. hBMVECs treated with A804598 (A80) critically protected the mitochondrial health and minimized the cytosolic escape of AIF-1 and Cytochrome *C*. Western blots from whole cell lysates, cytosolic fractions were normalized to β-actin and HSP60 for mitochondrial fractions (*n* = 3). One-way ANOVA was used for the statistical analyses, marked as * *p* ≤ 0.05, ** *p* ≤ 0.01, *** *p* ≤ 0.001, **** *p* ≤ 0.0001 and ns (not significant).

**Figure 6 antioxidants-11-01328-f006:**
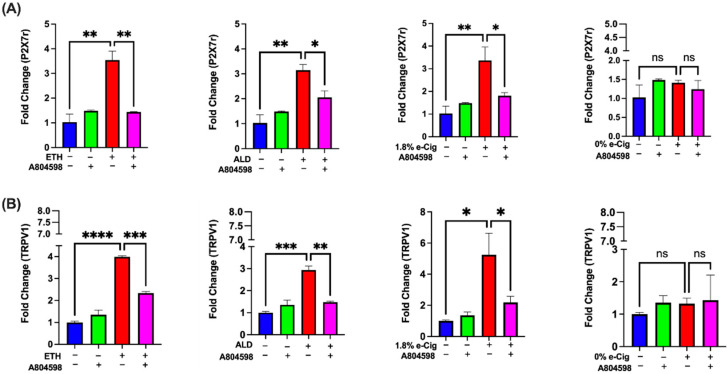
**ETH, ALD, and 1.8% e-Cig exposure increased the gene expression of P2X7r and TRPV1 channels in hBMVECs.** (**A**) hBMVECs treated with ETH, ALD, and 1.8% e-Cig exposure increased the P2X7r gene expression by four-fold while A804598 prevented such increases. (**B**) Simultaneously, expression of TRPV1 ion channel was also enhanced six-fold after exposure to ETH, ALD, or 1.8% e-Cig and A804598 treatment ameliorated the increases. q-PCR data were normalized with GAPDH (*n* = 3). One-way ANOVA was used for statistical analyses, marked as * *p* ≤ 0.05, ** *p* ≤ 0.01, *** *p* ≤ 0.001, **** *p* ≤ 0.0001 and ns (not significant).

**Figure 7 antioxidants-11-01328-f007:**
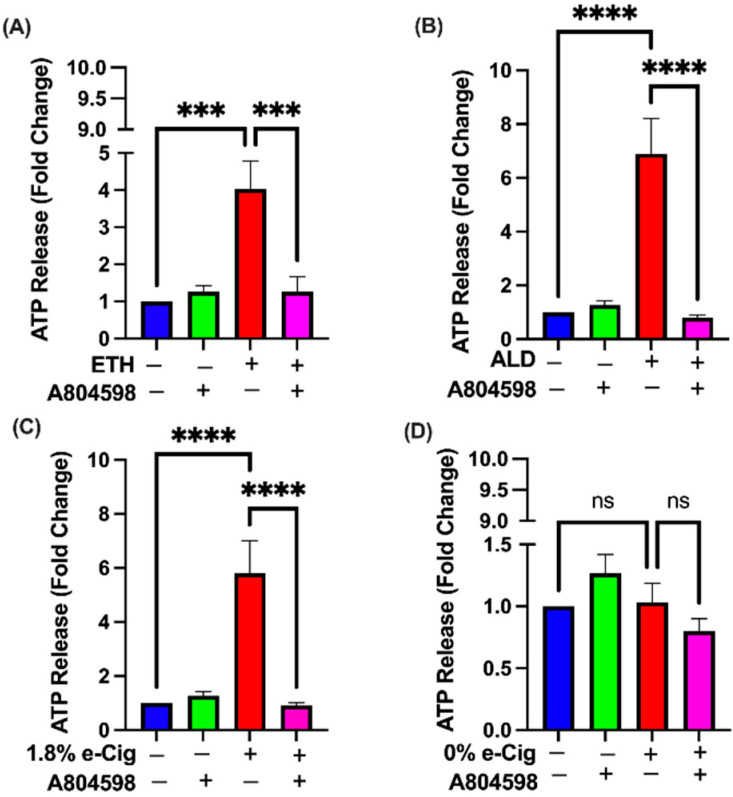
**extracellular ATP release stimulated by ETH, ALD, and 1.8 % e-Cig exposure in hBMVECs and A804598 blocked the eATP release.** hBMVECs exposed to (**A**) ETH presented a four-fold increase in eATP levels. Both (**B**) ALD and (**C**) 1.8% e-Cig demonstrated eight-fold increases in eATP, whereas (**D**) e-Cig with 0% nicotine showed no impact on eATP levels. Blockage of purinergic receptor (P2X7r) and TRPV1 ion channels by A804598 reduced the eATP levels (*n* = 3). One-way ANOVA was used for statistical data, where *** *p* ≤ 0.001, **** *p* ≤ 0.0001 and ns (non-significance).

**Figure 8 antioxidants-11-01328-f008:**
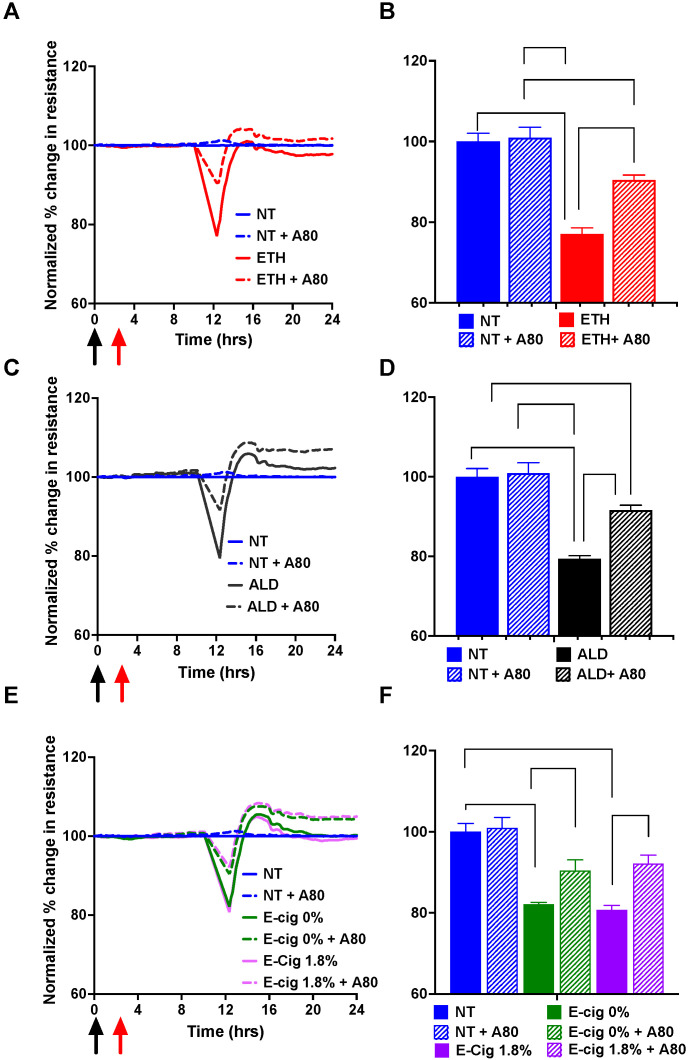
**P2X7r inhibition protected barrier function.** Trans-Endothelial Electrical Resistance (TEER) was measured continuously in hBMVECs with or without P2X7r inhibitor A804598 (at time 0 marked as black arrow) and exposed to ETH (**A**,**B**), ALD (**C**,**D**), or 1.8% e-Cig (**E**,**F**) (at 2 h marked as red arrow). After baseline TEER was achieved (between 6–7 days), the inhibitor and stimulators were added, and resistance was measured at 4000 Hz for the duration of the time shown. (**A**,**C**,**E**) are representative images from hBMVEC exposed to ETH, ALD, and 1.8% e-Cig respectively. (**B**,**D**,**F**) are average normalized percent change in TEER with positive SD (*n* = 3–5). Significance values from a two-way ANOVA analysis is shown in the graphs.

## Data Availability

Data is contained within the article and Appendix A.

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
