# Peer review of "Blocking of P2X7r Reduces Mitochondrial Stress Induced by Alcohol and Electronic Cigarette Exposure in Brain Microvascular Endothelial Cells"

_antioxidants, 2022, doi:10.3390/antiox11071328_

Round 1

Reviewer 1 Report

In this study, authors found that ethanol (ETH), acetaldehyde (ALD), and electronic cigarette (1.8% nicotine) conditioned media (e-Cig) induce mitochondrial dysfunction, oxidative stress, intracellular accumulation of calcium, endoplasmic reticulum stress, and trigger pro-apoptotic and pro-inflammatory pathways in cultured cells of a cerebral microvascular endothelial cell line. Moreover, these three agents affect tight junction properties in cultured endothelial cells. Authors found that inhibiting the P2X7 receptor, which has been involved in neuroinflammation and blood brain barrier dysfunction, prevents the deleterious effects of ETH, ALD, and e-Cig.

Overall, the manuscript is well organized and well written. The Methods section provides enough information to allow experimental replication. Results are well described and data are clearly presented (i.e., WB are convincing).

Below are several comments/suggestions that authors may want to address to increase the quality and significance of their study:

The use of ALD is not well justified in the manuscript.

Reference 6 is cited multiple times regarding the effects of ethanol in endothelial cells. However, the cited study was performed only in neurons.

Line 70. “By blocking P2X7r and TRPV1 ion-channels, P2X7r antagonist could protect mitochondrial health.” Emphasize that this is the authors’ hypothesis.

Introduce what spare respiratory capacity (SRC) and reserve respiratory capacity (RRC) are.

Fig. 1. It seems that A8 alone has a negative effect on oxygen consumption rates (OCR) and SRC. Apparently, differences between non-treated and A8 are significant, but statistics are missing. Please, add p-values. Also, discuss why A8 reduces OCR and how a combination of A8 and ETH, ALD or eCig complete restores OCR and SRC.

It is not clear the rationale of introducing SDH-II if there is no data related to it.

Why authors hypothesize that ETH/ALD/eCig could affect the protein levels of COX-IV and –II?

Line 243. “We analyzed expression of these two proteins using Western blot and found a 45 % drop in Complex- II expression and a 30% reduction in Complex-IV expression”. Between what groups?

Line 248. “Depletion of SDH and COX by ETH, ALD and 1.8% e-Cig conditioned media presumably reduced the spare respiration levels in hBMVECs” Better discuss this in the Discussion section.

Line 270. “ROS and proinflammatory signals triggered by ETH and cigarettes smoke are known to control the movement of Ca2+ through calcium-permeable ion channels including TRPV1 and P2X7 receptors [12, 30, 31].” Here there is no experimental or bibliographic evidence that ETH/eCig can induce ROS and inflammation in endothelial cells. Reference 12 is related to neurons, and references 30 and 31 seem not to be related to oxidative stress in endothelial cells. This makes the connection between Fig 2 and Fig 3 weak.

Figure legends should not present any description or interpretation of data, only a clear description of the experiments presented, for example, what technique was used, concentration and timing of treatments, number of replicates and independent experiments, etc.

Line 407. “To understand the role of addictive drugs on blood brain barrier (BBB) integrity and ATP release, we quantified extracellular ATP (eATP) levels in BMVECs stimulated with ETH, ALD and 1.8% e-Cig conditioned media.” According to the data presented in Fig 7, BBB integrity has not been measured. This should be omitted here and mentioned later in the section 3.8.

Fig 8. Assuming that all experiments were conducted with overnight treatments (~16 h after treatments) as indicated in the Method section, calcium impairment, AMP levels and pro-apoptotic signals occur 16 h after ETH/ALD/eCig treatments. However, in Fig 8, TEER is mostly affected by treatments 12 h after the insult, and restored 4 h later (16 h after treatment). Could authors discuss why TEER is restored at the time when other impairments occur?

Fig 9. Incorporating the findings of the study in the Fig 9 would help to understand the mechanism proposed by authors. Improving the quality of the figure is highly recommended (for example, replace a drawing of a purple row from an ATP to two ATPs).

Reviewer 2 Report

This is a study done in the D3 cell-line of brain microvascular endothelial cells. It impacts on impairment of blood-brain barnier permeability elicited by abuse drugs (i.e. alcohol, acetaldhyde and e-cigarette-nicotine). These stressors impair Ca2+-dependent mitochondrial and ER functions through alterations of purinergic P2X7 receptors, as revealed by their elevated gene expression. Authors find than P2X7r blockade wiht compound A804598 at 10µM, protects cells and mitochondria from the damage elicited by those stressors. In so doing, they argue that P2X7r may become a target to improve mitochondrial health and preserve the structure of the BBB against the above mentioned stressors.

Study design is fine, with multi-methodological approaches, and an adequate statistical analysis of results. From a pharmacological point of view, a dose-response curve with compound A804598 could add strenght to the study, at least with some parameter.

Reviewer 3 Report

The authors present an interesting study with regards to ‘lifestyle’ factors which have been shown to exert a great influence over the vasculature and its state of health. Specifically, the authors utilise a cell line of the brain endothelium and expose it to conditions whereby the cellular environment has fixed concentrations of alcohol or e-cigarette material. The mitochondrial response to said conditions was then measured, with alterations to mitochondrial dynamics observed. Downstream impact on additional cellular aspects such as barrier function are also reported, with a reduction in impedance observed, while the influence of P2X7 within this context highlighted with antagonists towards such reported to ameliorate the effects of alcohol/e-cigarettes.

In reviewing the manuscript, I had a number of concerns. The following should be considered by the authors when preparing a suitable revision.

1.       How was the effective concentration of the P2X7 antagonist determined?

2.       How were the concentrations of the stimulants (i.e. alcohol, e-cigarette fluid, etc) optimised for use in this study? Is there any translational value to the concentrations utilised?

3.       Were the primers utilised for the qPCR studies examined as to whether they were MIQE guidelines compliant?

4.       The authors utilise collagen coatings when performing the impedance measurements, however it is not indicated as to whether this was utilised in regular cell culture. Given that the extracellular matrix can impart functional changes to cells, can the authors clarify whether collagen was employed consistently throughout the study, or just for the impedance studies?

5.       In many of the western blots there are non-specific bands present. How can the authors be certain that the bands they are labelling as the target protein of interest are such?

6.       Similarly regarding the Westerns, in comparing the bands to the densitometry reported the trends at times do not quite appear to be what is displayed in the Western. The authors should revise the densitometry to ensure the values are what are presented in the graphs.

7.       While the authors utilise scatter plots for the most part in their data, it would be appreciated if the N-number was reported in each figure legend for clarity.    

8.       The formatting of the graphs and data is good for the most part, however in certain instances the labelling can be quite hard to read/is too small. For example, in Figure 8 there is ample space to increase the size of the labels/legends.

9.       It would be useful if in Figure 8 it was labelled when the stimulants were added and when they were removed in the graphs.

10.   In this reviewers experience utilising apparatus’ such as the one used for measuring the impedance of a cell monolayer in this study can see the measurements confounded by changes in the media composition. In other words, changes in the impedance can be caused by changes in the ionic profile of the media and not necessarily by changes in the cell monolayer. Can the authors clarify how they accounted for these changes to discern the changes made by the addition of the stimulants/antagonist from a physical perspective as opposed to changes in the cell monolayer?

11.   In Figure 1, the antagonist appears to exert its own effect on the respiration index, before then being shown to alleviate the effects of the stimulants (e.g. alcohol). Can the authors explain why the antagonist appears to have such polarising effects in this instance, but not quite in other indices measured?

Round 2

Reviewer 1 Report

Authors have addressed all the comments.

Reviewer 3 Report

The authors have suitably addressed my comments and the manuscript is much improved.